



# The European Preinstrumental Earthquake Catalogue EPICA, the 1000-1899 catalogue for the European Seismic Hazard Model 2020

Andrea Rovida, Andrea Antonucci, Mario Locati

Istituto Nazionale di Geofisica e Vulcanologia, Milano, 20133, Italy

*Correspondence to:* Andrea Rovida (andrea.rovida@ingv.it)

**Abstract.** The European PreInstrumental earthquake CAtalogue EPICA (Rovida and Antonucci, 2021; https://doi.org/10.13127/epica.1.1) is the 1000-1899 seismic catalogue compiled for the European Seismic Hazard Model 2020 (ESHM20), an outcome of the project Seismology and Earthquake Engineering Research Infrastructure Alliance for Europe (SERA), in the framework of the European Union's Horizon 2020 research and innovation programme. EPICA is the update of the SHARE European Earthquake Catalogue SHEEC 1000-1899, with which it

shares the main principles - mostly transparency, repeatability, and continent wide harmonization of data – as well as the compilation strategies and methods.

The version 1.1 of EPICA contains 5703 earthquakes with either maximum intensity ≥5 or Mw ≥4.0, with a spatial coverage from the Atlantic Ocean to the west to 32°E in longitude, and from the Mediterranean Sea to northernmost Europe.

EPICA relies upon the updated knowledge of European preinstrumental seismicity provided by the data gathered in the European Archive of Historical Earthquake Data AHEAD. Such data are both macroseismic intensity data supplied by descriptive historical seismological studies and online macroseismic databases, and parameters contained in regional catalogues. As done for the compilation of SHEEC 1000-1899, these datasets were thoroughly analysed in order to select the most representative of the knowledge of each earthquake, independently from national constraints.

Selected intensity distributions are processed with three methods to determine location and magnitude based on the attenuation of macroseismic intensity and are combined with parameters harmonised from modern regional catalogues.

This paper describes the compilation procedure of EPICA version 1.1, its input data, the assessment of the earthquake parameters and the resulting catalogue, which is finally compared with its previous version. Technical solutions for

accessing the catalogue, both as downloadable files and through webservices.

## 1 Introduction

Among their essential input datasets, seismic hazard evaluations require an accurate earthquake catalogue with a time coverage long enough to sample the temporal and spatial features of the seismicity of the study area. For this reason,

in areas with moderate and low seismicity, such as most of Europe, earthquake parameters assessed with instruments in the last decades have to be integrated with those of pre-instrumental earthquakes, which are usually derived from macroseismic observations. Parameters of macroseismic origin - magnitude in particular - have to be as consistent as possible both internally and with instrumental ones, and their reliability depends on the accuracy of the background historical research, and the approaches used for assessing them.

According to Rovida et al. (2020a), the availability and quality of preinstrumental earthquake data at the European scale is uneven because of the fragmentation of data repositories and their different features and levels of update. These authors also point out that only one third of the European earthquakes of the period 1000-1899 CE are supported by descriptive studies supplying or not macroseismic intensity data, whereas another third is known only through parametric catalogues. For the latter earthquakes it is usually very difficult, and sometimes impossible, to trace back

the historical information they rely upon. In addition, although the methodologies for assessing earthquake parameters from intensity data, each with its pros and cons, are nowadays robust, they rarely provide a unique solution from the same intensity dataset, and the selection of a single best method is not straightforward (Cecić et al., 1996; Bakun et al., 2011; Stucchi et al., 2013; Provost et al., 2022). This fragmentation and variety of both data and methods strongly affects the consistency of earthquake catalogues across country borders. For example, a very good representation of

the long-term seismicity at the national scale is provided by the catalogues of Switzerland (ECOS-09; Fäh et al., 2011), France (F-CAT17; Manchuel et al., 2018), and Italy (CPTI15; Rovida et al., 2020b; 2022). These three catalogues provide robust parameters derived with advanced, well-calibrated and well-documented methodologies exploiting the richest historical macroseismic databases in the world. However, the common earthquakes in the three catalogues occurred at the border of Switzerland, Italy and France in the western Alps, presents diverse magnitude estimations

due to the differences in both the input intensity distributions and the methods used for their processing. Such discrepancies of course have important consequences on the elaborations based on them such as, in the case of seismic hazard assessment, the computations of seismic activity rates, which may result inconsistent across country borders (e.g. Rong et al., 2011; Beauval et al., 2020; Provost et al., 2022).

In order to deal with this situation, the realization of the European Seismic Hazard Model ESHM13 (Wössner et al.,

2015) in the framework of the 2009-2013 project Seismic Hazard Harmonization in Europe (SHARE), included the compilation of an earthquake catalogue harmonised across national borders. In particular, the ESHM13 required a homogeneous catalogue compiled in terms of moment magnitude with transparent and repeatable procedures, based on the most updated knowledge provided by the results of previous European initiatives, and by regional contributions. A specific task of the project aimed at the compilation of the 1000-1899 part of the earthquake catalogue from scratch,

which resulted in the SHARE European Earthquake Catalogue (SHEEC) 1000-1899 (Stucchi et al., 2013; SHEEC 1000-1899 from now on). For earthquakes of the 20th century, a specific update (Grünthal et al., 2013) of the European-Mediterranean Earthquake Catalogue EMEC (Grünthal and Wahlström, 2012) was adopted.

Earth System
Science
Data

In the framework of the SERA (Seismology and Earthquake Engineering Research Infrastructure Alliance for Europe) 2017-2020 project, the creation of the European Seismic Hazard Model 2020 (ESHM20; Danciu et al., 2021), the new

release of ESHM13, required an update of all the input datasets, that were compiled almost ten years before (Basili et al., 2018). Among these databases, the new version of SHEEC 1000-1899 was compiled and named European PreInstrumental Earthquake CAtalogue - EPICA version 1.1 (Rovida and Antonucci, 2021).

The present paper describes the new catalogue, which takes into account the available updates in the input data, summarizing the procedures for their selection and parametrization. After providing an overview of the catalogue

content, with particular reference to the improvements with respect to its previous version, the availability and accessibility of the data is described.

## 2 Compilation procedure

The compilation of EPICA version 1.1 adopts the same procedures used for SHEEC 1000-1899, detailed in Stucchi et al. (2013) and summarised in Fig. 1. Previous efforts for the creation of a long-term continent-wide European

earthquake catalogue, e.g. the pioneering works of Kárník (1969, 1971), Van Gils (1988), and Van Gils and Leydecker (1991) and the more recent EMEC catalogue (Grünthal and Wahlström, 2012), consisted in the recompilation of national earthquake catalogues, and the conversion of their magnitudes to a common scale. To improve the cross-border homogeneity of the parameters, instead of collating national catalogues resulting from varied data and criteria, SHEEC 1000-1899 re-assessed earthquake locations and magnitudes from raw macroseismic intensity data

(Macroseismic Data Points, hereafter MDPs), selected from a unified and homogeneous database, with the same procedures throughout Europe.

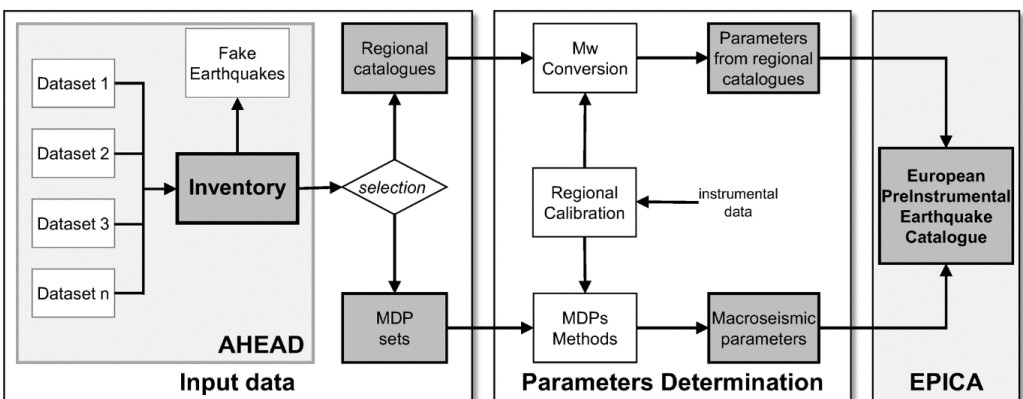

**Figure 1: Compilation procedure of EPICA version 1.1, adopted from SHEEC 1000-1899 (Stucchi et al., 2013).**

This approach followed for the first time the recommendations of Van Gils (1988) and Stucchi (1994), who recognised the limitations of merging national catalogues into a European one and suggested the creation of a unique database of primary data, both macroseismic intensity distributions and parametric catalogues, to serve as a base for the uniform



assessment of earthquake parameters. Several European joint efforts of the last 20 years pursued the collection, integration and publication of historical earthquake data sparse in local and regional archives in varied formats, and

the ultimate outcome is the European Archive of Historical Earthquake Data AHEAD (Albini et al., 2013; Locati et al., 2014; Rovida and Locati, 2015). AHEAD continuously collects, archives, organizes, and supplies data of different types and provenance on European earthquakes of the period 1000-1899 CE as derived from i) regional and national macroseismic databases, ii) seismological descriptive studies on specific earthquakes, periods, or areas, iii) major modern parametric catalogues. In this way, AHEAD presents the multiplicity of studies and datasets that may refer to

the same earthquake, providing diverse information. Datasets related to the same earthquake are thoroughly examined one by one, compared, and then associated.

SHEEC 1000-1899 exploited the wealth of data systematically inventoried in AHEAD analysing the datasets associated to each earthquake and selected among them the most representative of the knowledge of each earthquake, disregarding its national provenance. In the selection process, only published and/or publicly available datasets were

considered, and those providing MDPs had the highest priority. In the presence of multiple sets of MDPs for the same earthquake, the selection considered the characteristics of the research that produced them, preferring dataset accompanied by thorough descriptions of the performed investigation, the consulted historical sources and their seismological interpretation. In general, the most recent datasets are selected, unless they provide the very same MDPs as older although more comprehensive studies. In the absence of any MPDs sets for an earthquake, entries from

parametric catalogues were chosen as source of data for that earthquake. Also in these cases, preference was given to parametric catalogues accounting for their input information and the methods for assessing earthquake parameters. This procedure ensures the adoption of the most reliable and updated data, as well as the unequivocal identification of entries of different datasets related to the same earthquake, and earthquakes missing in one regional catalogue or another. In addition, catalogue entries related to fake events are easily recognised and sorted out.

Once a comprehensive and orderly list of earthquakes was made from AHEAD, the parameters of earthquakes with MDPs were assessed with three different methods : i) Boxer 4.0 (Gasperini et al., 1999; 2010), ii) MEEP (Musson and Jiménez, 2008), and BW (Bakun and Wentworth, 1997). These three methods rely on the attenuation of macroseismic intensity as a function of the earthquake magnitude and the distance of every MDP from the source, a feature that depends on regional attenuation characteristics and the peculiarities of intensity assessment. For each method, an

attenuation model was calibrated for five European regions, by means of the same set of calibrating events, i.e. earthquakes of known instrumental magnitude and with reliable MDPs distributions (Gomez Capera et al., 2015). For earthquakes without MDPs, the most reliable location available from regional catalogues was adopted, and magnitude was re-assessed from the conversion of epicentral intensity ($I_0$) with five regional empirical relations specifically derived from the same dataset used for calibrating the methods for processing MDPs, for the sake of homogeneity.

Parameters from MDPs and regional catalogues were then used to assess a set of final parameters. The final location consists in the location determined from MDPs whenever is available, or from regional catalogues otherwise. When Mw determinations from MDPs and from the regional catalogue are both available, the final Mw and related uncertainty is the weighted mean of them, otherwise, it is obtained either from MDPs or from regional catalogues.

Earth System
**Science**
**Data**
Open Access Discussions

## 3 EPICA features and content

Being the update of SHEEC 1000-1899 and adopting the same compilation procedures, EPICA version 1.1 had to take into account the innovations concerning both i) input macroseismic datasets and catalogues, and ii) regional calibrations of the methods for parameterizing MDPs, or new methods. In the last years AHEAD has been updated with European macroseismic datasets and regional catalogues that have been subject to some update and new publications, as detailed in the following. On the other hand, since the publication of SHEEC 1000-1899 in 2013, new

datasets on recent earthquakes that can significantly improve the used calibration datasets have not been published, and there were no major innovations to justify a revision of the regional calibrations. The only exception is the updated calibration of the Boxer method developed for the Italian catalogue CPTI15 (Rovida et al., 2020b). In addition, no robust new method for deriving location and magnitude from intensity data has been proposed in the literature.

In conclusion, the compilation of EPICA version 1.1 followed the same approach of SHEEC 1000-1899 described

above (Fig. 1), incorporating new input datasets provided by the updated content of AHEAD and revising many choices, whereas the strategies for the definition of earthquake locations and magnitudes are unaltered. This approach also complied with the time and financial resources allocated by the SERA project for the ESHM20, which were considerably lower than those available in SHARE, entirely dedicated to ESHM13.

EPICA version 1.1 contains 5703 earthquakes with either maximum intensity ≥5 or Mw ≥4.0, with a spatial coverage

from the Atlantic Ocean to the west to 32°E in longitude and from the Mediterranean Sea to northernmost Europe. It relies upon 160 sources of MDPs and 39 parametric catalogues.

### 3.1 Input data

AHEAD, the European Archive of Historical Earthquake Data is a dynamic repository, conceived to be continuously expanded and updated. In the period between the compilation of SHEEC 1000-1899 and EPICA, i.e. between 2012

and 2019, several new MDPs sets and updated earthquake catalogues were published. Among the regional nodes contributing data to AHEAD, the Italian Archive of Historical Earthquake Data (Rovida et al., 2017; from now on ASMI: Archivio Storico Macrosismico Italiano) and the French macroseismic database SisFrance (BRGM-EDF-IRSN/SisFrance, 2016) underwent significant updates. At the same time, the results of several investigations on specific earthquakes, areas and/or periods have become available in the scientific literature. Consequently, the contents

of AHEAD have been enriched with 81 sources of data not considered before, dealing with more than 6700 earthquakes, including 1488 new ones (Rovida et al., 2021). Out of the new data sources, 65 are macroseismic studies providing more than 273000 MDPs, and five are updated regional earthquake catalogues. In addition, several records derived from already considered sources were included, mostly because the minimum intensity considered in the archive was lowered. The current version of AHEAD, published online in May 2021, contains data on nearly 5800

earthquakes derived from 360 sources, and 8180 intensity datasets with 145500 MDPs in total. With respect to the version upon which SHEEC 1000-1899 was based, the number of earthquakes supported by MDPs increased from the 57% of the total to the 66%.

Following a thorough analysis of the content of AHEAD and the application of the criteria described above, with particular reference to new sources of data, a list of 5703 earthquakes supported by 160 sources of MDPs and 39

parametric catalogues (see https://www.emidius.eu/epica/data_sources.htm; last accessed 22 March 2022), was selected for the compilation of EPICA version 1.1. MDPs distributions are available for 3622 earthquakes. i.e. the 64% of those in the catalogue (Fig. 2). The selected sources of MDPs mainly consist of the regional nodes of AHEAD, namely the Italian Archive of Historical Earthquake Data ASMI, which in turn provides EPICA with 77 different sources of data, SisFrance, ECOS-09, the Greek Hellenic Macroseismic Database (HMDB.UoA; Kouskouna and

Sakkas, 2013) and Macroseismic Database of the Southern Balkan area (University of Thessaloniki, 2003), and the Spanish macroseismic database (Instituto Geografíco Nacional, 2010). In all, these regional databases provide the 78% of the MDPs, and the 92% of the earthquakes for which MDPs are available, i.e. the 59% of the earthquakes in EPICA. The remaining MDP sets derive from studies related to single earthquakes, small areas, or specific periods. As a result, the geographical distribution of earthquakes with MDPs is rather unbalanced towards central (UK, France,

Switzerland) and south-eastern (Italy, Greece) Europe (Fig. 2).

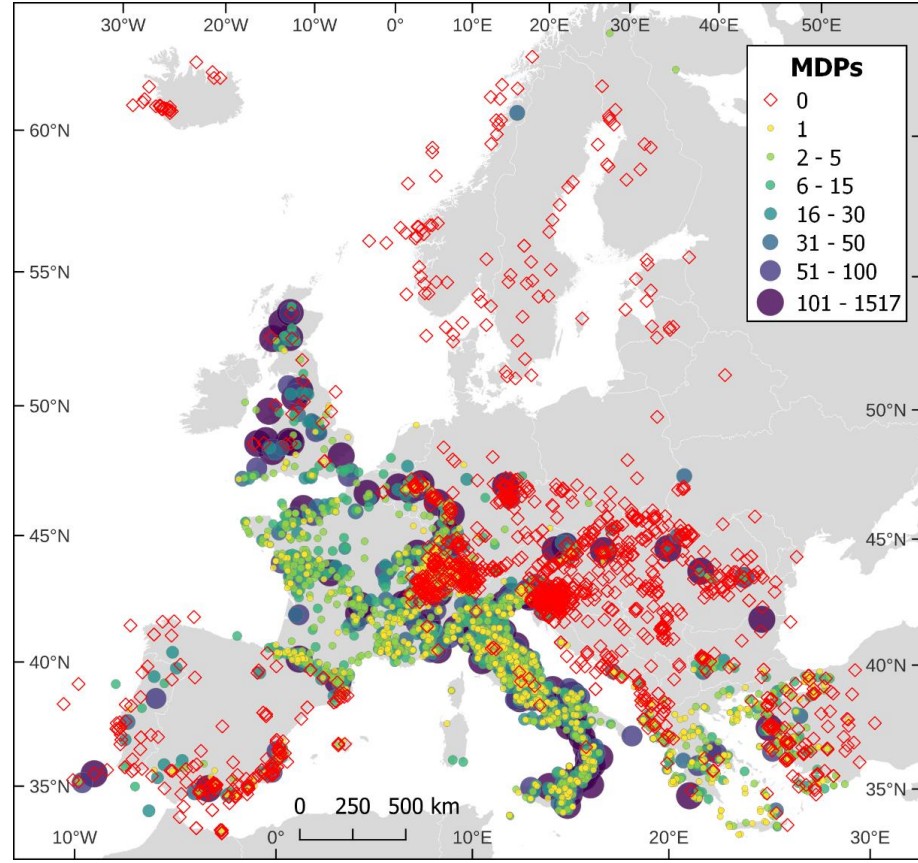

**Figure 2: Number of MDPs for each earthquake in EPICA, according to the selected dataset. MDPs = 0 means that MDPs are not available and a record from a parametric catalogue is selected.**

In all, sources of macroseismic data used in the compilation of EPICA derive from investigations published between

1975 and 2018, although most of them were published after the mid-1980s (Fig. 3a). The majority of the earthquakes with MDPs rely on dataset published after 2003 (Fig. 3b), when the Greek datasets of the University of Thessaloniki



(2003) and a study contributing to most of the Hellenic Macroseismic Database (Taxeidis, 2003) were published, together with some studies on Italian and Swiss earthquakes. Conversely, the largest number of MPDs was published starting from 2008, with several important contributions from Italy, Belgium, and the UK. Most of the MDPs date

between 2016 and 2018, thanks to the releases of the latest versions of SisFrance and of the Catalogue of Strong Italian Earthquakes (CFTI5med; Guidoboni et al., 2018), contributing 805 and 430 earthquakes respectively, although most of their content is the same of the previous versions. Out of the considered sources of MDPs, 36 studies providing MDPs to 2014 earthquakes are not among those used for SHEEC 1000-1899 mostly because they were not yet published or, in 10 cases, they were not considered although already available. These are mostly studies on Italian

earthquakes, because ASMI did not yet exist at that time and the Italian portion of AHEAD was less updated than the rest of Europe. EPICA also relies upon 39 regional catalogues (https://www.emidius.eu/epica/data_sources.htm; last accessed 22 March 2022), related to 5511 earthquakes, and selected as much as possible among those published and according to the transparency of their compilation procedures and their date of publication. The most recent catalogues satisfying such criteria and covering all the study area date as back as 1972, and nearly half of them is more than 20

years old (Fig. 3c), although they account for one fifth of the earthquakes (Fig. 3d). With few exceptions, these catalogues are mostly related to low seismicity areas, such as Northern and Eastern Europe and Portugal, and the majority of the earthquakes is reported in catalogues compiled in the last ten years. However, as pointed out in Rovida et al. (2020a), the data that recent catalogues rely upon might be much older.

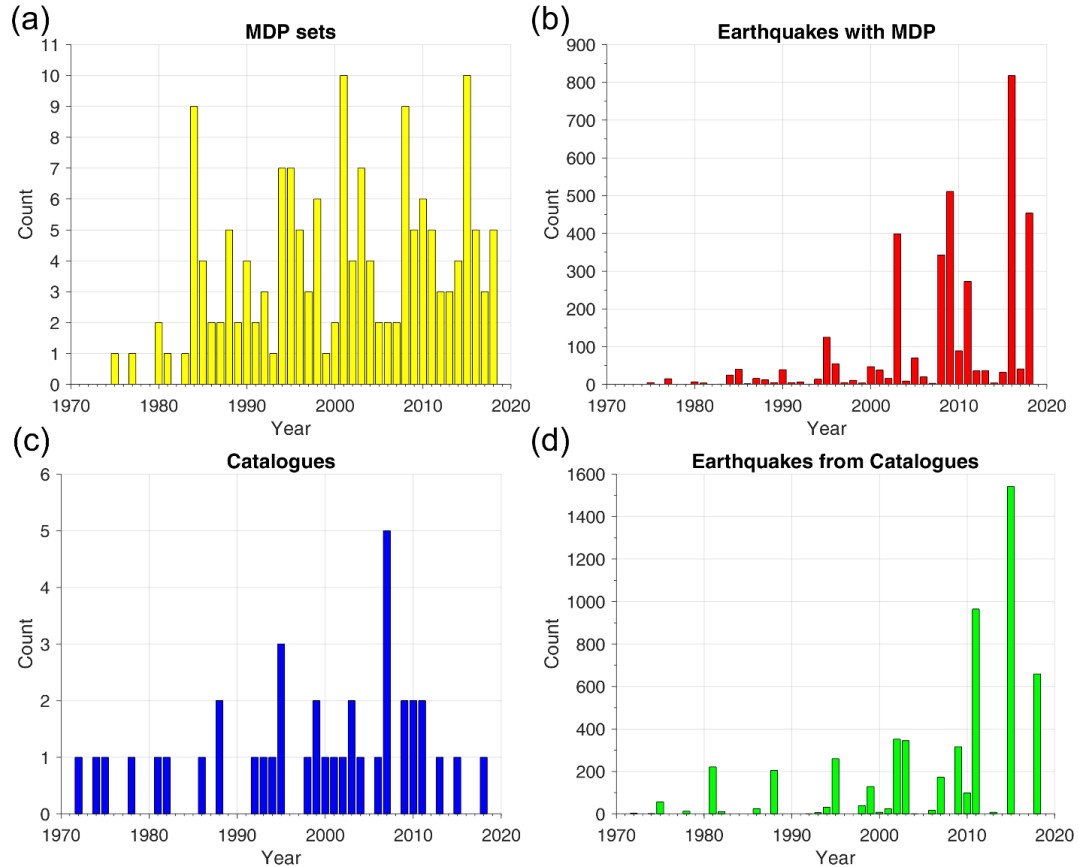

Figure 3: a) Number of data sources providing EPICA with MDPs distributions according to their year of publication and b) number of earthquakes with MDPs according to the year of publication of their source. c) Number of regional catalogues considered in EPICA according to their year of publication and d) number of earthquakes from regional catalogues according to the year of publication of the catalogue.

Following the selection principles of EPICA and in its period of interest (i.e. 1000-1899), only the four catalogues listed below have been published since 2012 and were considered in addition or substitution of those contributing to SHEEC 1000-1899:

- CPTI15 version 1.5 (Rovida et al., 2016; 2020b), contributing new parameters for 1541 Italian earthquakes;
- FCAT-17 (Manchuel et al., 2018), contributing new parameters for 659 French earthquakes;
- Hammerl and Lenhardt (2013), contributing new parameters for 8 earthquakes in Lower Austria;
- Leydecker (2011), contributing the same data as the previous version for 236 German earthquakes.

In conclusion, for 3445 of the 5073 earthquakes in EPICA both a MDP set and a parametric catalogue were selected, whereas for 2066 and 177 earthquakes respectively a record from a parametric catalogue or a MDP set were selected. It is worth noting that 145 earthquakes supported by MDPs and recent historical macroseismic investigations are not contained in any current or past earthquake catalogue. Conversely, AHEAD allowed EPICA to exclude about 180



earthquakes that, although present in the selected current earthquake catalogues, are assessed as fakes by published and consistent historical investigations.

## 3.2 Earthquake parameters

As mentioned in Section 2, parameters in EPICA were assessed following the same approach of SHEEC 1000-1899 (Stucchi et al, 2013; Gomez Capera et al., 2015), which foresaw the definition of two sets of parameters, each made of

epicentre location with uncertainty, epicentral intensity, and magnitude and releted uncertainty:

1)  assessed form MDPs with homogeneous and repeatable procedures
2)  derived from regional catalogues, coherently with those assessed from MDPs.

The combination of the two sets determines the final parameters, obtained from the selection of the location from either set 1) or 2), and assessing the magnitude as the weighted mean of those resulting from sets 1) and 2), depending on the

availability.

The catalogue file displays the final parameters and those derived as both sets 1) and 2).

### 3.2.1 Parameters from MDPs

In EPICA, parameters from MDPs, referred to as set 1) above, are determined for 3297 earthquakes out of the 3622 for which MDPs are available, because the intensity distributions of the remaining 325 earthquakes are too meager to obtain

robust parameters.

As in SHEEC 1000-1899, three methods for assessing macroseismic parameters were considered, i.e. Boxer (Gasperini et al., 1999; 2010), Meep (Musson and Jiménez, 2008) and BW (Bakun and Wentworth, 1997).

Attenuation models specific for each method were defined by means of the same sets of recent calibrating events in five European regions (Fig. 4) defined for SHEEC 1000-1899 (details are in Stucchi et al., 2013; Gomez Capera et al., 2015):

- Stable continental region (SCR)
- Western Alps and Pyrenees (WAP)
- Betics (BET)
- Apennines, North-Eastern Alps and Dinarides (APD)
- Broad Aegean, shallow (BAS)

Four additional areas were defined in SHEEC 1000-1899 for earthquakes in Iceland (ICE), offshore Portugal (TSZ), in the Aegean with intermediate depth (BAI), and Vrancea (VRD), although no attenuation models were defined because of the lack of intensity data and only catalogues were selected as input data in these areas (Fig. 4).

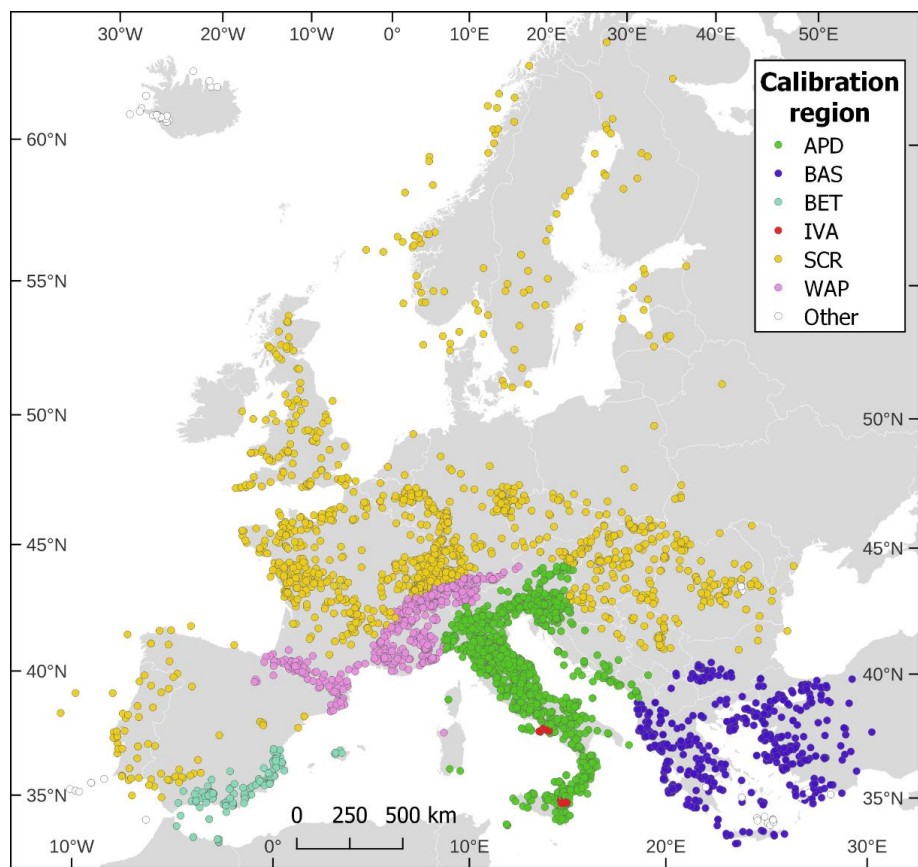

**Figure 4: Calibration regions defined for the assessment of EPICA's parameters. APD: Apennines, North-Eastern Alps and Dinarides; BAS: Broad Aegean, shallow; BET: Betics; IVA: Italian volcanic areas; SCR: stable continental region; WAP: Western Alps and Pyrenees.**

Because of the lack of new or updated macroseismic data on recent earthquakes in most of Europe, the regional calibrations of SHEEC 1000-1899 have been applied unaltered in EPICA, apart from the APD region. In this area, the calibration of Boxer realised from an improved and robust set of MDPs and instrumental magnitudes for the Italian catalogue CPTI15 (Rovida et al., 2020b) superseded the outdated one from the CPTI04 catalogue (CPTI Working Group, 2004) that was considered in SHEEC 1000-1899. In addition, a new region was defined for the Italian volcanic areas (IVA) of Mt. Etna and Vesuvius-Phlegrean Fields, where a new I0 to Mw conversion strategy developed for the Italian catalogue CPTI15 (Rovida et al., 2020b) was adopted. This strategy copes with the peculiarity of ground motion attenuation and the low energy of the earthquakes in that volcanic areas. It consists of the determination of local magnitude from epicentral intensity with the relation for the Etna area by Azzaro et al. (2011), and then the conversion of local magnitude to Mw with two relations specific for Etna (Tuvè et al., 2015) and Vesuvius-Phlegrean Fields (Petrosino et al., 2008) in the respective areas.





Taking into account the experience, the extensive tests and the choices made for compiling SHEEC 1000-1899, the
selected method was Boxer in most cases, with MEEP and BW used as exceptions respectively for the events in the UK
and for a few offshore events. The summary of the parameters assessed with the three methods in the five regions is
shown in Table 1.

**Table 1: Earthquakes with MDPs in each calibration region and method used for assessing parameters. APD: Apennines,
North-Eastern Alps and Dinarides; BAS: Broad Aegean, shallow; BET: Betics; IVA: Italian volcanic areas; SCR: stable**
**continental region; WAP: Western Alps and Pyrenees.**

| Method | APD | BAS | BET | SCR | TSZ | WAP | Total |
|--------|-----|-----|-----|-----|-----|-----|-------|
| Boxer | 1349 | 357 | 52 | 767 | | 662 | 3185 |
| BW | | 26 | | 1 | 1 | | 30 |
| MEEP | | | | 82 | | | 82 |
| Total | 1349 | 383 | 52 | 850 | 1 | 662 | 3297 |

Empirical relations between epicentral intensity I0 and Mw (Table 2) are derived from the same datasets used for
calibrating the three methods. Also in this case, the relations used in SHEEC 1000-1899 were applied in EPICA, apart
in the APD area.

**Table 2: Empirical relations between epicentral intensity I0 and Mw defined in each calibration region from Stucchi et al.
(2013) except that of APD, from Rovida et al. (2020b). APD: Apennines, North-Eastern Alps and Dinarides; BAS: Broad
Aegean, shallow; BET: Betics; SCR: stable continental region; WAP: Western Alps and Pyrenees.**

| Region | Equation | σ |
|--------|----------|---|
| BET | Mw = 1.487 + 0.552 * I0 | 0.38 |
| SCR | Mw = 0.528 + 0.655 * I0 | 0.25 |
| WAP | Mw = 1.441 + 0.502 * I0 | 0.31 |
| APD | Mw = 1.827 + 0.467 * I0 | 0.11 |
| BAS | Mw = 3.404 + 0.355 * I0 | 0.32 |

### 3.2.2 Parameters from regional catalogues

The regional catalogues selected from AHEAD provide parameters (location and epicentral intensity or
magnitude, or both) for 5251 earthquakes. In EPICA, locations are adopted from the catalogues without
any modifications, whereas the magnitude is determined according to the same procedure adopted in
SHEEC 1000-1899. In detail, as summarised in Table 3, Mw originally provided by catalogues is adopted
without any modifications, otherwise it is re-assessed from epicentral intensity according to the conversion
relations in Table 2 and the region each earthquake belongs to (Table 4). In few remaining cases, only Ms
or ML are supplied by the catalogue, and they are converted to Mw according to the relations of Bungum
et al. (2003) and Grünthal et al. (2009), respectively. As an extreme case, four catalogues do not specify
the adopted type of magnitude, and in the lack of any further information, it is assumed to be equivalent to
Mw.






**Table 3: Number of earthquakes from regional catalogues considered in EPICA according to their calibration region and the origin of EPICA magnitude. APD: Apennines, North-Eastern Alps and Dinarides; BAS: Broad Aegean, shallow; BET: Betics; SCR: stable continental region; WAP: Western Alps and Pyrenees.**

| Mw Origin | APD | BAS | BET | SCR | WAP | Other areas | TOTAL |
|---|---|---|---|---|---|---|---|
| Original Mw | 1277 | 316 | 13 | 763 | 756 | 191 | 3316 |
| From I0 | 465 | 235 | 254 | 804 | 73 | | 1831 |
| From Ms | | | | 38 | | | 38 |
| From Ml | | | | 18 | | | 18 |
| Unspecified | | | | 23 | | 25 | 48 |
| Total | 1742 | 551 | 267 | 1646 | 829 | 216 | 5251 |

**Table 4: Catalogues considered in EPICA and magnitude determination strategy.**

| Catalogue | Mw | From I0 | Ms | ML | Unspec. |
|---|---|---|---|---|---|
| CPTI15 (Rovida et al., 2016; 2020a) | 1444 | | | | |
| ECOS-09 (Fäh et al., 2011) | 711 | | | | |
| FCAT-17 (Manchuel et al., 2018) | 656 | | | | |
| Papazachos and Papazachou (2003) | 332 | | | | |
| Musson and Sargeant (2007) | 67 | | | | |
| Oncescu et al. (1999) | 64 | 58 | | | |
| Martinez Solares and Mezcua Rodriguez (2002) | 19 | 327 | | | 1 |
| Olivera et al. (2006) | 8 | 9 | | | |
| EMEC (Grünthal and Wahlström, 2012) | 8 | | | | |
| Vilanova and Fonseca (2007) | 4 | | | | |
| Kondorskaya and Ulomov (1999) | 1 | | | | |
| Martinez Solares and Lopez Arroyo (2004) | 1 | | | | |
| Pelaez et al. (2007) | 1 | | | | |
| Živčić (2009) | | 308 | | | |
| Leydecker (2011) | | 236 | | | |
| Zsìros et al. (1988) | | 186 | | | |
| Herak (1995) | | 158 | | | |
| Soysal et al. (1981) | | 152 | | | |
| ZAMG (2010) | | 86 | | | |
| Labak and Broucek (1995) | | 70 | | | |
| Sulstarova and Kociu (1975) | | 56 | | | |
| University of Helsinki (2007) | | 49 | 38 | | |
| Shebalin and Leydecker (1998) | | 37 | | | |
| LNEC (1986) | | 20 | | | 4 |
| Meidow (1995) | | 15 | | | |
| Grigorova et al. (1978) | | 13 | | | |
| Observatoire Royal de Belgique (2010) | | 12 | | | |
| Kondorskaya and Shebalin (1982) | | 11 | | | |
| Grünthal (1988) | | 8 | | | |
| Hammerl and Lenhardt (2013) | | 7 | | | |
| Boborikin et al. (1993) | | 6 | | | |
| Pagaczewski (1972) | | 4 | | | |
| Shebalin et al. (1974) | | 2 | | | |
| Nikonov (1992) | | 1 | | | |
| Musson (1994) | | | | 18 | |
| Martins and Mendes Victor (2001) | | | | | 22 |
| Icelandic Meteorological Office (2007) | | | | | 13 |
| Ambraseys and Sigbjörnsson (2000) | | | | | 8 |
| Total | 3316 | 1831 | 38 | 18 | 48 |


EPICA provides magnitude uncertainty for all the earthquakes. Among the considered regional catalogues, ECOS-09, F-CAT17, CPTI15, and Papazachos and Papazachou (2003) provide magnitude uncertainties, which are reported in EPICA, whereas for the other catalogues uncertainties are those associated to the conversion relation from I0, otherwise they are assumed as equal to 0.3 or 0.5.



### 3.2.3 Final parameters


Taking into account the selected datasets and the two sets of parameters assessed as described above, the final parameters in EPICA are determined as described in the following, where the names of the respective fields in the catalogues file are indicated in parentheses.

*Origin Time (Year, Mo, Da, Ho, Mi)*

The time of occurrence of each earthquake derives from the selected dataset.

*Location and uncertainty (Lat, Lon, TEpi, LatUnc, LonUnc, TEpiUnc)*

The epicentral location determined from MDPs, i.e. Set 1) as described above, is always preferred to the location proposed by regional catalogues (Set 2), and the latter is adopted only for earthquakes without MDPs or when MDPs are not parametrised.

As a result, 3297 (57.8%) epicentres in EPICA are from MDPs, and 2257 (39.6%) from the selected regional catalogues. In addition, 149 epicentres (2.6%) relate to earthquakes for which the available data do not allow a robust determination of the location, and are marked as "preliminary". Among the locations from MDPs, 3187 are from Boxer, 82 from MEEP, and 28 from BW (Fig. 5).

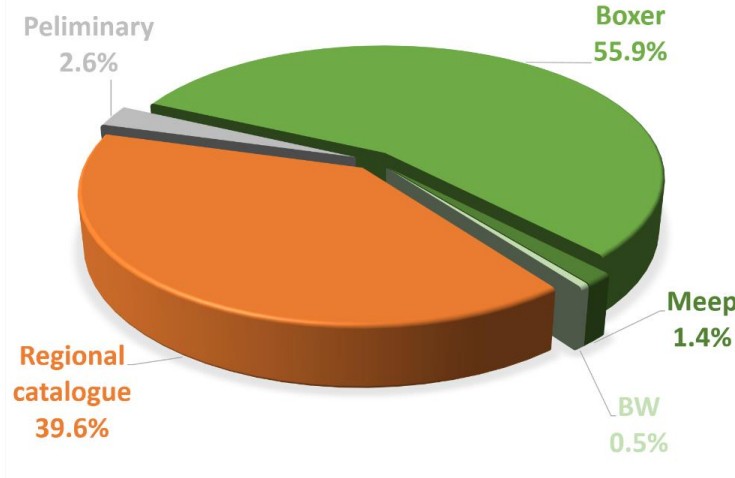


**Figure 5: Types of epicentre locations in EPICA.**

The uncertainty associated to epicentral locations from Set 1) are determined for 1944 earthquakes with the relevant method, namely 1869 with Boxer and 75 with Meep. Both Boxer and Meep calculate epicentral uncertainty only with enough data, while BW does not. When the uncertainties are not assessed, a default value of 30 km is assigned to epicentres of onshore earthquakes, and of 50 km for offshore ones. For locations in Set 2), the uncertainty of epicentral

location is taken from the selected catalogues (854 earthquakes) when provided and expressed in km, otherwise it is converted into km from degrees (116 earthquakes). Default values of 39.9 (for onshore earthquakes), 49.9 (for offshore earthquakes), or 99.9 km (when the catalogue proposes "undefined" uncertainty or values/classes >50 km) are adopted for 1158 earthquakes for which the selected catalogue does not provide epicentral uncertainty. Location uncertainty is not provided for 278 earthquakes.



*Depth (H, Hunc, TH)*

The assessment of the focal depth of historical earthquakes is controversial and affected by high uncertainties (e.g. Gasperini et al., 2010). For this reason, depth is provided for 738 earthquakes, only, and is mostly (635 earthquakes) derived from regional catalogues. Hypocentral depth from MDPs distributions is available only for the 82 earthquakes located with Meep, because neither Boxer nor BW calculate it. Meep also provides depth uncertainty for 25

earthquakes.

The areas defined as BAI and VRD in the field "Reg" of the catalogue file respectively indicate intermediate and deep earthquakes of the Aegean and Vrancea regions, although the hypocentral depth is not always expressed.

*Magnitude (Mw, MwUnc, TMw)*

As in SHEEC 1000-1899, magnitude is determined as a combination of the values obtained from the MDPs processing

and the selected regional catalogue, when they are both available. In these cases, Mw is assessed as the weighted mean of the two values, with arbitrary weights of 0.75 and 0.25 attributed to the Mw from MPDs and from the regional catalogue, respectively. In continuity with SHEEC 1000-1899, reverse weights are given to ECOS-09 and CPTI15. When only one of the two sets of parameters is available, the respective Mw is adopted.

Mw is determined as the weighted mean for 3127 earthquakes (55% of the total), it derives from the regional catalogue

for 2124 (37%) earthquakes and from MDPs for 170 (3%) of them (Fig. 6). In addition, Mw is not determined for 282 earthquakes because the data they rely upon are not robust enough.

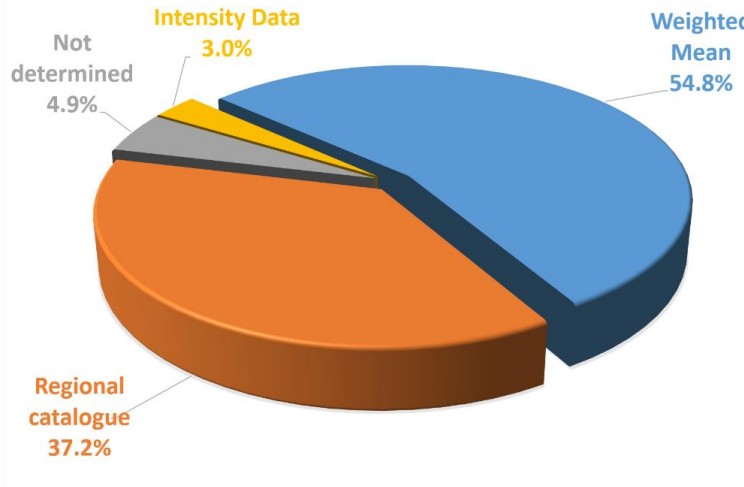

**Figure 6: Types of magnitudes in EPICA.**

All the Mw estimates in EPICA are accompanied with their uncertainties, determined as follows:

• for Mw determined from MDPs, the uncertainty provided by the selected method is adopted if larger than 0.3, otherwise it has been fixed at 0.3; BW assesses magnitude uncertainty as a function of the number of MDPs used, Meep uses a bootstrap resampling technique and Boxer computes both formal and bootstrap uncertainties.

• for Mw obtained from regional catalogues the adopted uncertainty is either: i) the uncertainty provided by the catalogue when available; ii) a default value of 0.3 when Mw is obtained from the conversion of I0 or another



type of magnitude or is not available in the source catalogue; iii) a default value of 0.5 when the type of the
       original magnitude is not expressed;

- for Mw obtained as the weighted mean of the values from MDPs and from regional catalogues, uncertainty is
  calculated as the square root of the sum of the squares of the uncertainties, each multiplied by its own assigned
  weight.

Figure 7 shows the geographical distribution of the earthquakes according to the way Mw is determined.

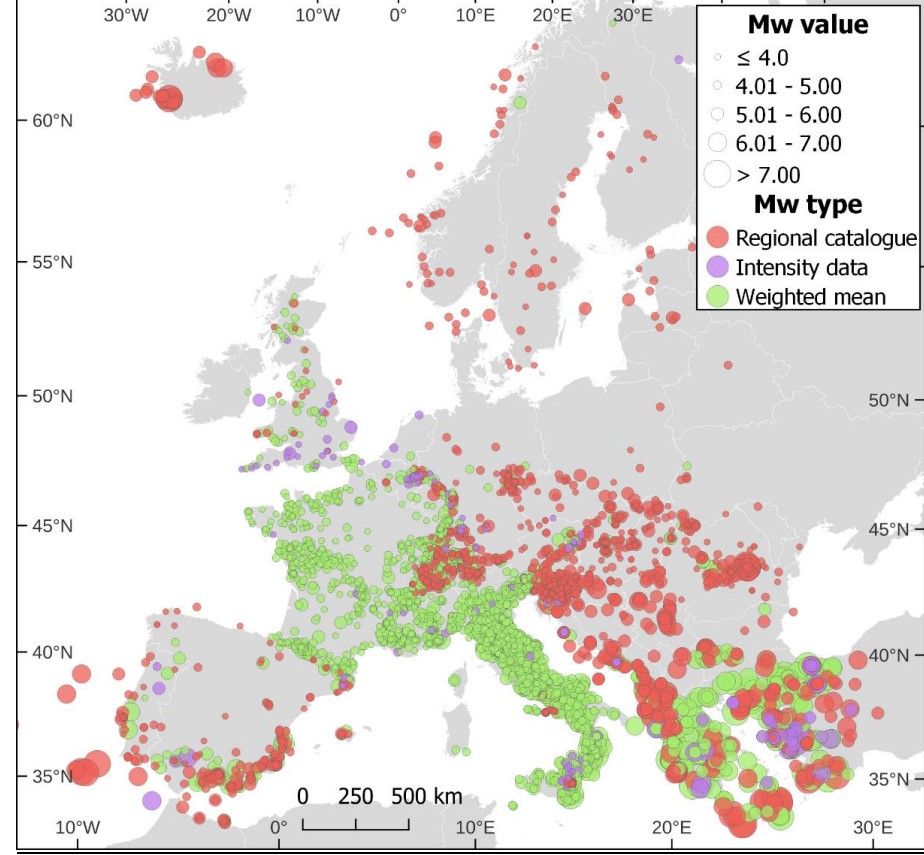

**Figure 7: Geographical distribution of earthquakes in EPICA and their type of final magnitude.**

## 4 Data availability

EPICA version 1.1 (Rovida and Antonucci, 2021) is available at https://doi.org/10.13127/epica.1.1 and is released

under a Creative Commons Attribution 4.0 International (CC BY 4.0) license.

The catalogue file is downloadable in both OpenDocument Spreadsheet Document (.ods) MS Excel (.xlsx) formats.
The downloadable spreadsheets contain all catalogue parameters, as described in the previous sections. The
description of the file is shown in a dedicated webpage (https://emidius.eu/epica/description_table_fields.htm; last
accessed 22 March 2022) and contained in a separate sheet of the downloadable files.





EPICA version 1.1 is also available through AHEAD's webservices according to the standards of the International
Federation of Digital Seismograph Networks (fdsnws-event), and the Open Geospatial Consortium, in particular the
Web Feature Service (OGC WFS), and the Web Map Service (OGC WMS). As shown in Table 5, the three standards
may provide the catalogue encoded in different output formats to meet the users' needs. The documentation for these
web services is available at https://www.emidius.eu/AHEAD/services/ (last accessed 22 March 2022).

**Table 5: Standards, outputs and output formats of the webservices for accessing EPICA.**

| WS standard | Output | Output formats |
|---|---|---|
| fdsnws-event | EPICA (preferred) origins and magnitudes and all alternatives from AHEAD | QuakeML 1.2 (XML), CSV (text), GeoJSON |
| OGC WMS | Styled map with EPICA origins and magnitudes | PNG, JPG, GIF, PDF, GeoTiff |
| OGC WFS | Geographical features of EPICA origins and magnitudes | GML 3.2, GML 3.1, GML 2, KML, Shapefile, GeoJSON, CSV, MS Excel |

The OGC WFS standard returns the complete set of EPICA parameters, whereas the fdsnws-event standard cannot
include any macroseismic information (in the case of EPICA, the origin and number of intensity data, the maximum
reported intensity, and so on). On the other hand, the fdsnws-event standard allows the user to obtain the earthquake
origins and magnitudes from all the alternative catalogues archived in AHEAD, together with the (preferred) solution
of EPICA.

EPICA data can be downloaded and displayed in the widely used open source GIS software QGIS using the QQuake
plugin (Locati et al., 2021), that gives access to various type of seismological data - such as parameters, macroseismic
intensity data, seismic stations, or seismogenic faults - via a set of pre-configured webservices. QQuake allows also
to download the input macroseismic data, obtained from the AHEAD macroseismic web services that works similarly
to fdsnws-event but instead of providing QuakeML 1.2, implements the macroseismic package of QuakeML 2.0
(Euchner and Kästli, 2014; Locati, 2014; Euchner et al., 2016; Kästli and Euchner 2018).

Finally, EPICA is also included in AHEAD as the main catalogue, and is the reference earthquake catalogue for the
period 1000-1899 in the EPOS Integrated Core Service portal (https://www.ics-c.epos-eu.org/; last accessed 22 March
380    2022).

## 5 Comparison with SHEEC 1000-1899

The updates and variations in the input datasets described in the previous sections reflect in several differences
between EPICA and SHEEC 1000-1899.

Earthquakes in EPICA are 5703 instead of the 4722 in SHEEC 1000-1899. The reasons for such an increase are the
introduction of 46 studies providing MDPs and four catalogues that are not among those used for compiling SHEEC
1000-1899, mostly because they are more recent or, at a lesser extent, they were already published but not considered.
The latter are mostly studies on Italian earthquakes, because ASMI did not yet exist and the Italian portion of AHEAD
was less updated than the rest of Europe.





In EPICA there are 4668 earthquakes in common with SHEEC 1000-1899, and the records related to half (2332) of
them are the same as in SHEEC 1000-1899 (Table 6), i.e. they derive from the same sources of data, either MDP sets
(63) or catalogues (1559) or both (710), and thus present the same parameters.

**Table 6. Comparison between SHEEC 1000-1899 and EPICA contents.**

| Earthquakes | Same dataset as SHEEC | | | New dataset | | | TOTAL |
|---|---|---|---|---|---|---|---|
| | MDP | CAT | MDP+CAT | MDP | CAT | MDP+CAT | |
| Same | 63 | 1559 | 710 | - | - | - | 2332 |
| Modified | 360 | 10 | 3 | 141 | 513 | 1309 | 2336 |
| Added | 27 | 286 | 26 | 73 | 133 | 490 | 1035 |
| Total | 450 | 1855 | 739 | 214 | 646 | 1799 | 5703 |

On the other hand, EPICA includes 1035 earthquakes that were not in SHEEC 1000-1899 (Table 6). The majority
(696) of these earthquakes derive from 23 descriptive studies and three parametric catalogues published after the
compilation of SHEEC 1000-1899, which provide parameters from MDPs to 73 earthquakes, from catalogues to 133
earthquakes, and from both to 490 earthquakes. The remaining 339 earthquakes added to EPICA (Table 6) are included
from studies or catalogues already considered in SHEEC 1000-1899 because of the intensity/magnitude threshold
(319) lowered to intensity 5, or because of the revision of the datasets selection (20 cases). In addition, 2336
earthquakes already listed in SHEEC 1000-1899 are included in EPICA with a different set of data, in most of the
cases (1963) because such datasets are new, otherwise the reference dataset has changed to be consistent with other
similar choices (Table 6).

Figure 8 shows the locations of the earthquakes in EPICA and the comparison of their input datasets with respect to
SHEEC 1000-1899, fakes in SHEEC 1000-1899 are also mapped.

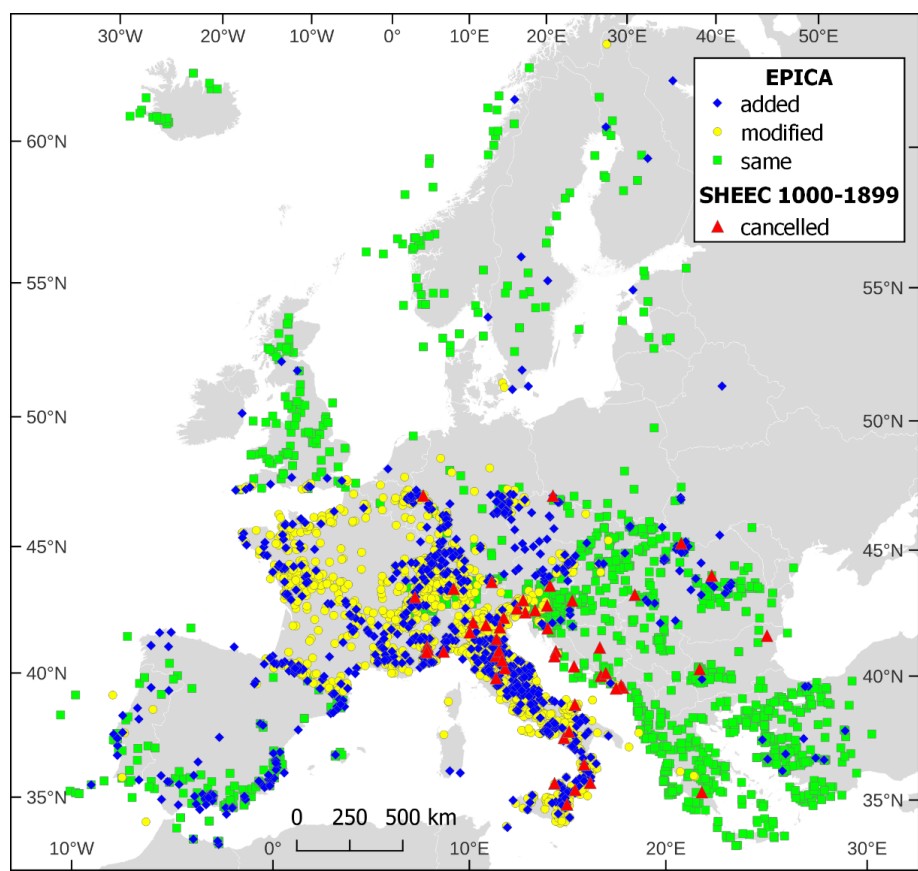

**Figure 8: Locations of the earthquakes in EPICA and comparison of their input datasets with respect to SHEEC 1000-1899. Earthquakes in SHEEC 1000-1899 that were cancelled are also shown.**

With respect to SHEEC 1000-1899, 49 earthquakes (Appendix A; Fig. 8) are not included in EPICA for the following

reasons:

- 42 of them have been assessed as fakes in a new study;

- 4 of them were duplications of other earthquakes;

- 3 of them were in the Swiss catalogue ECOS-02 (Swiss Seismological Service, 2002) but are not included in the updated version ECOS-09 (Fäh et al., 2011) without any motivation.

In addition to newly introduced earthquakes, new sources of data and the revision of the selection of the input datasets, impact on the parameters of 2336 earthquakes (see Table 6). In particular, the parameters of 501 earthquakes rely on new or different set of MDPs, those of 523 on a new or different regional catalogue, and those of 1312 on both different MDP sets and catalogues. As a whole, the earthquakes with parameters based on the homogeneous and reproducible processing of MDPs increased from 2447 to 3622 in EPICA, with a total of 49852 considered MDPs instead of 42581.

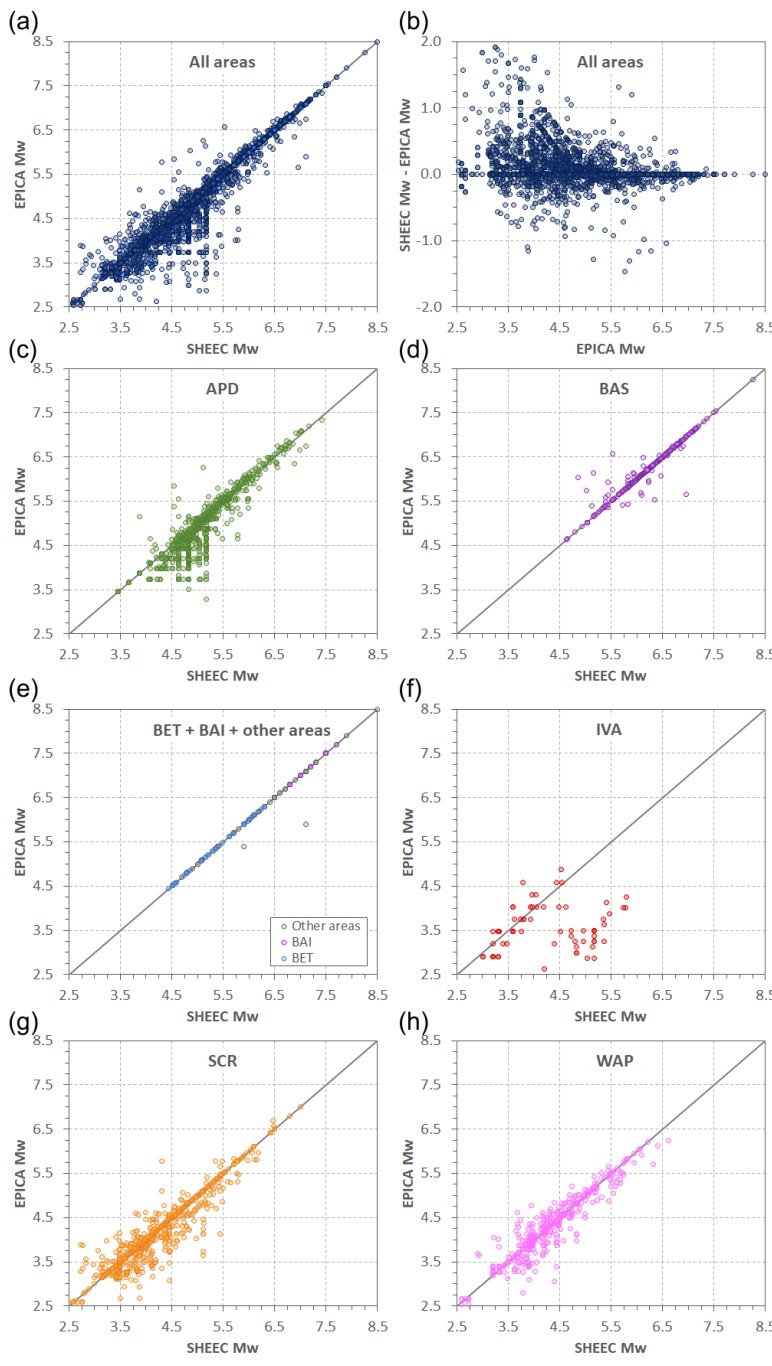


**Figure 9: Comparison between magnitude values in SHEEC 1000-1899 and EPICA for all the earthquakes (a) and (b), and in each calibration region (c to h).**



Figure 9 shows a comparison of the magnitudes of the 4668 earthquakes in both EPICA and SHEEC 1000-1899. In general, variations are mostly (93%) within ±0.5 Mw units, i.e. within the mean associated uncertainty which is 0.47 in EPICA. An overall decrease of the magnitudes values, up to 2.0 Mw units is observed in EPICA (Figs. 9a and 9b) and mostly affects the lowest magnitudes. The reason for such a decrease is the introduction of the new calibration of the Italian volcanic areas (IVA region; Fig. 9f) and, to a lesser extent, to the new calibration in the Apennine-Dinarides region (APD; Fig. 9c). However, for earthquakes in Italy, the magnitude decrease is mostly due to the consistent amount of new macroseismic data today available (see also Fig. 8, and Rovida et al., 2020b). In addition, the new F-CAT17 catalogue for France accounts for the variations in the SCR (Stable Continental region; Fig. 9g) and WAP (Western Alps – Pyrenees; Fig. 9h) regions. Some of the changes in the WAP and SCR regions are also the effect of the parameterization of MDPs from ECOS-09 that, for time constraints, was not fully done in SHEEC 1000-1899 and Mw from ECOS-09 catalogue were adopted. The few variations in the Aegean area (BAS;Fig. 9d), are mainly due to the correction of a few entries form the Turkish catalogue (Soysal et al., 1981). No variations are introduced in the BET and BAI regions, and, aside very few revisions, to other areas (Fig. 9e).

Some variations in the earthquake locations were also introduced. Excluding differences due to coordinate rounding, they relate to 1628 earthquakes, with an average distance of 16 km and a maximum of 678 km. The latter very high difference is referred to the 28 May 1897 earthquake, for which newly assessed MDPs on the coasts of Southern Italy (Molin et al., 2008) complemented those in Greece, and resulted in a more robust location at sea. However, only for 334 earthquakes the distance from the previous location is significant, i.e. ≥ 20 km. The latter differences are the consequence of the adoption of a new input dataset for 292 earthquakes, but also 42 corrections of compilation errors of SHEEC 1000-1899 that affected entries from the Slovenian (Živcic, 2009) and Turkish (Soysal et al., 1981) catalogues. In particular, the variations in the epicentres of earthquakes in Switzerland derive from the already mentioned parametrization of MDPs form ECOS-09. Similarly, for French earthquakes the epicentres are calculated from MDPs, while in SHEEC 1000-1899 were adopted from SisFrance 2010.

## 6 Conclusions

EPICA makes the best possible use of the available knowledge of European historical seismicity, as represented in the European Archive of Historical Earthquake Data AHEAD, to maximize the harmonization of earthquake data and parameters across country borders.

Although the procedures for deriving earthquake parameters are the same as in SHEEC 1000-1899, the wealth of new input data causes significant variations with respect to it, emphasizing the influence of quality macroseismic data derived from thorough historical investigations on earthquake location and magnitude. The increase in the number of earthquakes with MDPs indicates an overall improvement of the knowledge of the earthquakes, being the MDPs distributions the results of modern, thorough historical investigation of primary sources of information, which in turn reflects on the reliability of earthquake parameters that can be assessed with objective, homogeneous, and repeatable procedures (see Rovida et al., 2020a). Recent guidelines for implementing European data infrastructures, supported by several recent international initiatives such as the European Plate Observing System EPOS (Haslinger et al., 2022),





greatly contributed to improve the findability and accessibility of input data for compiling EPICA, achieving what was initially envisioned in Van Gils (1988) and Stucchi (1994).

However, the availability and reliability of studies and data across Europe is incomplete and inconsistent, with large differences from one area to the other. In particular, the location of earthquakes with MDPs is not homogeneously distributed, with a substantial lack of them in Northern and Central-Eastern Europe, and the level of update of the input datasets, both macroseismic studies and parametric catalogues, differs considerably.

In addition, the methods for the definition of macroseismic parameters and their associated uncertainties are

unchanged in the last decade and need to be improved.

Such a heterogeneity reflects on the uniformity of any Europe-wide catalogue, and the room for improving the knowledge of the European seismic history is still very large. Unfortunately, despite the widespread perception of the utmost importance of historical macroseismic investigations and data for the knowledge of long-term seismicity, the dedicated resources and manpower have been steadily decreasing in recent years in all countries and institutions. In

spite of this, AHEAD will continue to collect newly produced data independently of their amount, and EPICA will hopefully be updated on a regular basis, regardless of the requests of specific seismic hazard assessment projects, to incorporate any innovation in terms of both input data and parameterization procedures, and possibly expanded to neighbouring areas and to recent times.





**Appendix A**

Table A1: Records from SHEEC 1000-1899 not reported in EPICA according to the motivation indicated in the column "Note".

| En | Year | Mo | Da | Ho | Mi | Ax | Lat | Lon | Mw | Note |
|---|---|---|---|---|---|---|---|---|---|---|
| 10860 | 1178 | 04 | 15 | | | [Zadar] | 44.200 | 15.100 | 5.14 | fake in Alexandre and Alexandre (2012) |
| 10930 | 1183 | 12 | | | | Verona | 45.438 | 10.994 | 4.93 | fake in Guidoboni et al. (2018) |
| 11020 | 1197 | | | | | Brescia | 45.550 | 10.220 | 5.03 | fake in Guidoboni et al. (2018) |
| 11290 | 1223 | | | | | Gargano | 41.874 | 15.981 | 5.84 | fake in Camassi et al. 2008 |
| 11310 | 1223 | 01 | 08 | | | Transylvania | 46.500 | 25.000 | | fake in Alexandre and Alexandre (2012) |
| 11890 | 1268 | 11 | 04 | | | Trevigiano | 45.735 | 12.079 | 5.36 | fake in Camassi et al. 2012 |
| 12150 | 1279 | 04 | 24 | 17 | | Cividale del Friuli | 45.930 | 13.400 | 5.37 | fake in Camassi et al. 2012 |
| 12360 | 1287 | 04 | 11 | | | Cremona | 45.136 | 10.024 | | fake in Guidoboni et al. (2018) |
| 13010 | 1310 | | | | | Villa S. Giovanni | 38.250 | 15.667 | 5.17 | fake in Molin et al. (2008) |
| 13270 | 1323 | | | | | [Novi Vinodolski] | 45.200 | 14.700 | 5.99 | fake in Alexandre and Alexandre (2012) |
| 13580 | 1343 | 06 | 30 | | | Dalmatia | 44.000 | 15.000 | 5.99 | fake in Alexandre and Alexandre (2012) |
| 13720 | 1346 | 02 | 22 | 11 | | Ferrara | 44.836 | 11.618 | 5.31 | fake in Camassi and Castelli (2013) |
| 15090 | 1386 | | | | | Bosnia | 44.200 | 17.700 | 5.99 | fake in Alexandre and Alexandre (2012) |
| 16970 | 1444 | 08 | 04 | | | Szeged | 46.250 | 20.150 | | fake in Alexandre and Alexandre (2012) |
| 16990 | 1444 | 11 | | | | Eastern Bulgaria | 43.500 | 27.500 | | fake in Alexandre and Alexandre (2012) |
| 17150 | 1450 | | | | | 82 Balk.reg. | 42.700 | 23.300 | | fake in Alexandre and Alexandre (2012) |
| 17280 | 1455 | 02 | 03 | 20 | | Spilimbergo | 46.110 | 12.899 | | fake in Guidoboni et al. (2018) |
| 17530 | 1459 | 05 | 20 | | | Northern Croatia | 46.300 | 16.300 | 6.42 | fake in Alexandre and Alexandre (2012) |
| 17540 | 1461 | 06 | | | | Castelcivita | 40.500 | 15.250 | 5.17 | fake in Molin et al. (2008) |
| 18040 | 1473 | 01 | 20 | | | [Opuzen] | 43.000 | 17.600 | 5.57 | fake in Alexandre and Alexandre (2012) |
| 18230 | 1479 | 10 | 20 | | | Dalmatia | 43.000 | 17.600 | 5.99 | fake in Alexandre and Alexandre (2012) |
| 18260 | 1480 | 10 | 18 | | | [Stolac] | 43.100 | 17.900 | 5.57 | fake in Alexandre and Alexandre (2012) |
| 18890 | 1496 | 01 | 23 | 17 | | Central Dalmatia | 43.500 | 16.100 | 5.99 | fake in Alexandre and Alexandre (2012) |
| 19170 | 1502 | 09 | 23 | | | Cuneo | 44.500 | 7.500 | 5.17 | fake in Guidoboni et al. (2018) |
| 19890 | 1511 | 06 | 26 | | | Idrija | 46.000 | 14.000 | 4.09 | DUPLICATED |
| 19900 | 1511 | 06 | 26 | | | Idrija | 46.000 | 14.000 | 4.09 | DUPLICATED |
| 21870 | 1549 | 05 | 03 | | | Savona | 44.307 | 8.480 | 4.96 | fake in Camassi et al. (2015) |
| 22280 | 1556 | 01 | 24 | | | Illiria | 47.000 | 15.000 | | fake in Hammerl and Lenhardt (2013) |
| 22420 | 1559 | 01 | 24 | | | [Kotor] | 42.400 | 18.800 | | fake in Albini and Rovida 2018 |
| 23590 | 1571 | 11 | 01 | | | Innsbruck | 47.270 | 11.390 | 4.96 | fake in Hammerl (2015) |
| 25950 | 1600 | | | | | Palazzuolo | 44.113 | 11.548 | 5.36 | fake in Castelli et al. (1996) |
| 35448 | 1631 | 02 | | | | [Boka Kotorska] | 42.500 | 18.700 | 5.14 | fake in Albini and Rovida (2018) |
| 35525 | 1632 | | | | | [Boka Kotorska] | 42.400 | 18.400 | 5.99 | fake in Albini and Rovida (2018) |
| 40785 | 1669 | | | | | Piémont | 44.381 | 7.538 | 3.95 | fake in Camassi et al. (2015) |
| 43300 | 1687 | | | | | Castel Bolognese | 44.333 | 11.750 | 4.83 | fake in Molin et al. (2008) |
| 45100 | 1691 | 07 | 14 | | | Bovolenta | 45.333 | 11.833 | 4.83 | fake in Molin et al. (2008) |
| 63410 | 1751 | 07 | 31 | | | Karkonosze Mts. | 50.800 | 15.600 | 5.11 | fake in Leydecker (2011) |
| 64900 | 1755 | 10 | | | | Echallens,Lausanne | 46.580 | 6.650 | | NOT IN ECOS-09 |
| 69510 | 1769 | 09 | 25 | | | [Glarus] | 46.980 | 9.020 | | NOT IN ECOS-09 |
| 70700 | 1772 | 10 | 01 | | | Glarus | 46.980 | 9.020 | | NOT IN ECOS-09 |
| 75200 | 1780 | 01 | 03 | | | Monte Oliveto | 43.175 | 11.545 | | fake in Camassi et al. (2011) |
| 100800 | 1831 | 04 | 09 | | | Stilo | 38.500 | 16.500 | 5.03 | fake in Molin et al. (2008) |
| 106800 | 1839 | 08 | 18 | 01 | | Cosenza | 39.300 | 16.250 | 4.83 | fake in Molin et al. (2008) |
| 108152 | 1841 | 10 | 24 | 14 | 08 | Köln | 50.900 | 6.900 | 5.11 | fake in Lehmann and Leydecker (2014) |
| 119900 | 1858 | 08 | 06 | 12 | 15 | Ricigliano | 40.750 | 15.550 | 5.17 | fake in Molin et al. (2008) |
| 132600 | 1877 | 01 | 25 | 03 | 53 | Valbruna | 46.450 | 13.300 | 5.03 | DUPLICATED |
| 151500 | 1889 | 06 | 30 | 21 | 15 | Basso Tirreno | 38.583 | 14.583 | 5.17 | fake in Molin et al. (2008) |
| 159368 | 1894 | 10 | 07 | 02 | | Carpathians | 48.100 | 23.500 | 4.79 | DUPLICATED |
| 167900 | 1898 | 02 | 17 | 06 | 02 | S. Sofia | 43.917 | 11.917 | 4.83 | fake in Molin et al. (2008) |



## Author contribution

AR and AA curated the dataset, ML developed the web services for accessing the dataset. The three authors contributed to the manuscript and approved the final version.

## Competing interests

The authors declare that they have no conflict of interest.

## Acknowledgements

We are indebted with the Authors of SHEEC 1000-1899, in particular Massimiliano Stucchi, for their major and ground-breaking effort in harmonizing earthquake data across Europe, and to the countless researchers who contributed through time to the writing of the seismic history of Europe. We are grateful to our colleague Paola Albini for her precious contributions and constant support. We also thank the countless colleagues, contributors, and regional experts who provided us with their expertise, valuable feedback and suggestions. We wish to thank the ESHM2020 Core Group, in particular Laurentiu Danciu and the coordinators of the SERA WP 25, Fabrice Cotton and Stefan Weimer, and of the SERA Project, Domenico Giardini, for encouraging this effort and making it possible.
The European Union's Horizon 2020 Research and Innovation Programme provided founding under grant agreement No.730900.





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
