# Peer review of "The European Preinstrumental Earthquake Catalogue EPICA, the 1000-1899 catalogue for the European Seismic Hazard Model 2020"

_Earth System Science Data, 2022_

## Author Response (AR1)

**essd-2022-103: point-by-point response to referee comments and changes in the revised manuscript**

**RC1**

*1] I find it interesting for the reader to know the total number of multiple records contained in the AHEAD database.*

**We will add the suggested numbers in Section 3.1.**

**Done (line 164-166 in the revised MS).**

*2] The earthquakes studied on a macroseismic basis in the Italian area guide the entire process of creating European catalogs.*

*In Fig. 2 it is clear that in the Italian area there are many earthquakes with a number of MDps≥101. But in Fig. 2 it is not possible to understand as they are covered by the symbols with fewer MDps.*

*To better explain this there are two solutions: make an enlargement of Italy - or alternatively a table with the number of earthquakes in the various classes of MDps.*

**We will improve the readability of Figure 2 as suggested also in RC3. We will add a zoomed map, as Figure 2B, showing all the earthquakes with more than 100 MDPs, that are concentrated in the Italian region but present also elsewhere. We will also display in the legend of the two maps the number of earthquakes in each class of MDPs.**

**Done. We changed the symbol for catalogues and the color scale; we added the number of earthquakes n each MDPs classes and provide a zoom on earthquakes with 100+ MDPs**

*3] I suggest making a bar chart for magnitude classes (bin data) from the EPICA catalog. This would help the user to understand the real content of the catalog itself.*

**We will include the suggested chart as Figure 8, at the end of section 3.2.**

**Figure 8 added**

**RC2**

*I only noticed a slight inconsistency in the spelling of MEEP/Meep.*

**Thank you, we will fix it**

**Changed in the revised MS and in Figure 5.**

*I cannot judge the quality of the large dataset in whole because I don't know the details of the situation in other regions. I will focus only on a small part close to me - the Czech Republic and its immediate surroundings. The previous SHEEC catalogue contained only about 50 earthquakes for this low seismicity region, mainly from the German-Czech border area. In the EPICA catalogue, about 50 more earthquakes have been added, mostly taken from Leydecker 2011. Authors had no other choice – at the time of compilation of the catalogue there was no better source. But a few years ago, a revision of earthquakes with foci within and around the Czech Republic began. The first, albeit still very imperfect, result of this revision (PrachaÅ™., Pazdírková J., 2022, Historical Earthquake Database for the Bohemian Massif, https://doi.org/10.48790/HPAB-H834) was only made publicly available at the end of April 2022, one month after the EPICA manuscript was submitted. It turns out that some of the events listed in EPICA are false earthquakes, e.g. Iglau 1329, Cheb 1505, Northern Moravia 1786, Riesengebirge 1789, Trutnov 1882 and some others. Similar imperfections are likely to be found in other regions, depending on the level of exploration in particular area, and cannot be completely ruled out. However, this is why it is necessary to create an updated catalogue from time to time to reflect new research results.*

**We thank the Reviewer for pointing this new dataset out. In general, there is necessarily a time gap between the release of an earthquake catalogue and the publication of the input data it relies upon. EPICA was released to the public in April 2021 (see the dataset metadata). As stated in Section 3.1, as an input dataset of the European Seismic Hazard Model 2020, its compilation was completed in 2019, and obviously includes data published up to then. This is why in the Conclusion we state that the next version will hopefully be updated on a regular basis, independently from the time constraints of other projects.**

**RC3**

*Specific comments:*

*The language is generally good and easy to follow, but there are some cases of very long or unnecessarily complex sentences. I give some examples under "Technical corrections".*

**We will revise the text according to the suggestions.**

**Done.**

*Figure and table captions are generally very short and provide only the most basic information. To help "superficial readers", I would suggest adding a bit more detail to the captions. The authors should also make sure that all figure content and abbreviations are explained in the captions.*

**We will revise the captions, adding details and the missing explanations.**

**Done.**

*It is difficult to see the data in Fig. 2 with the symbology chosen. One options could be to use a less prominent symbol for events with 0 MDP, for example a small, red dot.*

**We will improve the readability of Figure 2, taking also into account RC1's suggestions.**

**Done (see response to RC1)**

*The paragraph describing Fig. 3 (p.6-7) is difficult to follow. Adding a short introduction, framing the description of the time distribution of sources, may help. Also, the difference between Fig 3a and 3b should be clearly explained. There are several grammatical errors (is/are) in that paragraph.*

**We will revise and improve the paragraph according to the suggestions.**

**Done (line 183-205). We also made the caption of Figure 3 more clear and change te title of the chart horizontal axis.**

*On page 8, line 214 it is stated that about 180 earthquakes from various catalogues have been identified as fakes. I suggest adding a list of those events (either by expanding the appendix or in an electronic supplement), as I think this list would be of interest to the wider community.*

**The mentioned earthquakes are among those classified as fakes in AHEAD. They are easily retrievable querying the online database and we do not think it is worth reporting them in the paper. Appendix A lists the earthquakes in SHEEC 1000-1899 that are not included in EPICA because they turned out to be fakes.**

*The first paragraph on page 11 describes to what extent the three macroseismic location procedures have been used. It would be interesting to include some general information on what has driven the choice of method in different cases.*

**We will add the requested information.**

**Done (lines 271-275).**

*Table 2 lists the relations between Mw and I0 used for each calibration region. I would suggest also including the attenuation models used for each region and method.*

> **The Boxer method actually uses one attenuation equation, in the functional form of Sibol et al. (1987), for each intensity class. This results in several different equations for each of the calibration regions, which are all published in Gomez Capera et al (2015), except that for Apennines and Dinarides region, published in Rovida et al (2020). We think that including them in the paper would be complicate and redundant, but we will make a clearer reference to Gomez Capera et al (2015) and Rovida et al (2020) in the revised MS.**

> > **Reference to to Gomez Capera et al (2015) and Rovida et al (2020b) added at lines 243 and 260.**

*The description of origin times on page 13 is very short. How are origin times derived from MDPs? And what is done if origin times are available both from MDP and catalog?*

> **We will expand and improve the description of origin times as suggested**

> > **Done.**

*On page 14, line 331, it is explained that the weights used in the magnitude determination are arbitrary. That is in principle ok, but it would be good to include some explanation of why those values were considered better than e.g. a 50/50 weighting. Also, please explain the term "reverse weights".*

> **The approach followed in the compilation of both EPICA and SHEEC 1000-1899 is conceived to exploit macroseismic data points at their best, because i) they are "codified" results of thorough investigation of historical information and ii) they can be objectively and homogeneously used to derive earthquake parameters. This reason is also behind the choice of the weights for the magnitudes. We realized that this concept deserves to be more highlighted in the MS, and we thank the reviewer for pointing it out.**

> > **The paragraph on the weights has been expanded in the revised MS.**

*Technical corrections:*

*Page 1, line 18: Coming from "northernmost Europe" myself, I find this term very unspecific – to me, for example, that would also include the Svalbard area. Whereas this is certainly a very minor issue, I would encourage the authors to use a different term.*

> **Ok.**

> > **Done.**

*Page 1, line 29: The last sentence in the paragraph seems to be incomplete.*

> **Yes it is, we will fix it.**

> > **Done.**

*Page 4, line 94: "sparse" seems to be misplaced.*

**Ok.**

**It seems ok to us**

*Page 5, line 131: "innovations" does not seem to be the most appropriate word here. Use "new developments" instead?*

**Ok.**

**Innovations" changed to "new developments".**

*Page 5, lines 156-160: As I read these sentences, new datasets contribute 273000 MDP, whereas the final dataset contains 145500 MDP. How is that possible?*

**We will double check and correct the numbers.**

**We corrected the number of MDPs contributed by new studies**

*On pages 5 and 6 there are several uses of the phrasing "the XX % of …". Here "the" should be removed.*

**Ok.**

**Corrected.**

*Page 8, lines 211-212: As I read this sentence, 3445 of the 5073 events in EPICA has both MDP and a catalogue listing, 2066 have only a catalogue listing and 177 only have MDP. These numbers do not sum up, please clarify.*

**We will double check and correct the numbers.**

**Corrected**

*Page 11, line 279: This sentence gives the impression that Table 4 gives information on the regions. Should be rephrased.*

**We will switch the positions of Tables 3 and 4, so that the actual Table 4 will be cited at the beginning of Section 3.2.2, and the actual table 3 where regions are mentioned.**

**Done.**

*Page 18, line 415: The sentence starting "In addition to…" is unclear.*

**Ok, we will rephrase it.**

**Done.**

*Page 20, line 428: It is stated that "…the magnitude decrease is mostly due to the consistent amount of new macroseismic data today available…". Please explain why this leads to a magnitude decrease.*

**Ok, we will clarify.**

**Done.**

*Page 20, lines 452-456: This is a very long and complex sentence. Splitting it up would improve readability.*

**Ok, we will split the sentence.**

**Done.**